# Bounded Context Management for
# Tabular Foundation Models on Stream Learning

**Jinmo Lee** [1]  **Doyun Choi** [1]  **Moongi Choi** [2]  **Jaemin Yoo** [1]

## Abstract

Tabular stream learning requires predictions on sequentially arriving examples under distribution shift. While standard methods adapt by updating model states, tabular foundation models (TFMs) make predictions conditioned on a labeled context in an in-context manner, making them a natural alternative for stream learning. This shifts the challenge from how to update the model to how to manage the context. We propose a future-information view that yields three practical requirements for context management: preserve recent examples, retain uncertain examples, and remove redundant examples. We instantiate these requirements as CURE (Context management via Uncertainty-aware admission and Redundancy-aware Eviction), a context-managing policy with entropy-gated admission and redundancy-aware eviction. Across seven streams, CURE shows up to 27.0% relative improvement over classical stream learners, remains robust across multiple TFM backbones, and ranks first among other policy variants. Code and datasets are available here.

## 1. Introduction

Tabular stream learning studies supervised prediction from sequentially arriving examples under bounded memory, real-time response requirements, and potential distribution shift (Aggarwal, 2007; Nguyen et al., 2015; Korycki & Krawczyk, 2022; Gama et al., 2014). Standard stream learners address this through online or incremental model-state adaptation, such as updating tree statistics or ensemble members (Domingos & Hulten, 2000; Gomes et al., 2017; 2019).

[1]Department of Computer Science and Engineering, Seoul National University, Seoul, Republic of Korea [2]The Kim Jaechul Graduate School of AI, KAIST, Daejeon, Republic of Korea. Correspondence to: Jinmo Lee <jinmo.lee@snu.ac.kr>, Jaemin Yoo <jaeminyoo@snu.ac.kr>.

*Proceedings of the 2nd ICML Workshop on Foundation Models for Structured Data*, Seoul, South Korea. 2026. Copyright 2026 by the author(s).

Recent tabular foundation models (TFMs) offer a different paradigm for tabular prediction. Given a labeled context $D$ and a query $x$, TFMs directly output a posterior predictive distribution $q_\theta(\cdot \mid x, D)$ without dataset-specific model updates (Hollmann et al., 2022; 2025). This in-context mechanism makes TFMs appealing for data streams since a model can adapt by changing the retained context (Lourenço et al., 2025). Therefore, the core challenge shifts from how to update the model to how to manage a context under bounded memory. A recent method, DualFIFO (Lourenço et al., 2026), has shown that first-in-first-out context updates can make TFMs competitive on streams, but it remains unclear which past examples should be retained for future queries.

Motivated by this gap, we introduce a *future-information view* that measures how much information a context provides to near-future queries. This yields three requirements for a context management policy: preserve recent examples to approximate the near-future distribution, retain uncertain examples with high potential label information, and remove redundant examples with overlapping evidence.

We implement these requirements as CURE (Context management via Uncertainty-aware admission and Redundancy-aware Eviction), a context update policy with entropy-gated admission and redundancy-aware eviction. Across seven streams, CURE achieves the best prequential accuracy over classical stream-learning baselines, improving by up to +19.59 points, shows consistent gains across multiple TFM backbones, and ranks first among controlled policy variants from the same design space.

## 2. Problem Setup

In this paper, we consider stream classification under the test-then-train prequential protocol (Gama et al., 2009; 2013). At each step $t$, a query $x_t$ arrives and the learner must predict before the true label $y_t$ is observed. After $y_t$ is revealed, the new labeled example $z_t = (x_t, y_t)$ becomes available for future predictions.

When a TFM is applied on data streams, the model outputs $q_\theta(\cdot \mid x_t, D_t)$ given a query $x_t$ and bounded context $D_t$ of previously observed feature-label pairs, while $\theta$ is fixed throughout the stream. Thus, adaptation to evolving data

can occur only by changing the retained context $D_t$.

We define this task as *bounded context management*. Given a pretrained TFM, a prequential stream $\{(x_t, y_t)\}_{t \geq 1}$, and a context budget $B$, the goal is to design an online update policy $\mathcal{U}$ that selects the next context using only information available up to time $t$:

$$D_{t+1} = \mathcal{U}(D_t, z_t), \qquad |D_{t+1}| \leq B. \tag{1}$$

As stated earlier, one notable prior method Dual-FIFO (Lourenço et al., 2026) shows that TFMs can be effective stream learners, but this does not provide an explicit criterion for the utility of retained examples. This motivates the future-information view introduced in Section 3.

# 3. A Future-Information View of Bounded Context

A key question in bounded context management is deciding which examples should be kept and which should be removed. We formalize this using the *near-future information* provided by a context. All supporting assumptions, lemmas, and theorems are stated and proved in Appendix A.

**Future-Information View.** Fix a stream step $t$ and a short horizon length $h$. Let $\mathcal{H}_t^+ = \{t+1, \ldots, t+h\}$ and define the near-future feature distribution as $\mathcal{P}_{t,X}^+ = \frac{1}{h} \sum_{s \in \mathcal{H}_t^+} \mathcal{P}_{s,X}$, where $\mathcal{P}_{s,X}$ is the feature marginal at step $s$. For a near-future query $x' \sim \mathcal{P}_{t,X}^+$, let $Y_{x'}$ be its label random variable.

**Definition 3.1** (Future usefulness of a bounded context). For a current context $D_t$, we define its future usefulness as

$$\mathcal{J}_t(D_t) = \mathbb{E}_{x' \sim \mathcal{P}_{t,X}^+} \left[ I(D_t; Y_{x'} \mid x') \right]. \tag{2}$$

Here, $I$ denotes pointwise mutual information evaluated at each $x'$. This says that a context is useful if it provides information about near-future query labels. However, it cannot be used as an online policy since $\mathcal{P}_{t,X}^+$ is unknown, and $\mathcal{J}_t$ is for the whole context rather than individual samples. We therefore decompose it into three practical signals.

**Recency.** The outer expectation in Eq. (2) depends on the unknown near-future feature distribution. Since recent examples often reflect the current data-generating concept (Losing et al., 2017; Chen et al., 2020), we use the recent window as a proxy for near-future query regions. Assumption A.1 and Lemma A.2 further formalize this, and show that the information measured on recent examples approximates the near-future information up to a distributional discrepancy term. This motivates reserving part of the context budget for recent examples.

**Uncertainty.** Recency tells us where future queries may appear, but not which examples are worth keeping beyond the recent window. For a newly observed example

$z_t = (x_t, y_t)$, the ideal admission criterion is its marginal contribution

$$\Delta_t(z_t \mid D_t) = \mathcal{J}_t(D_t \cup \{z_t\}) - \mathcal{J}_t(D_t). \tag{3}$$

This asks how much adding $z_t$ increases the future information of the context. Lemma A.3 rewrites this as

$$\Delta_t(z_t \mid D_t) = \mathbb{E}_{x' \sim \mathcal{P}_{t,X}^+} \left[ I(Y_{x_t}; Y_{x'} \mid x_t, x', D_t) \right], \tag{4}$$

meaning that $z_t$ is useful when its label helps predict future labels. Moreover, Theorem A.6 shows that, under Assumptions A.4 and A.5, prediction-time uncertainty $H(Y_{x_t} \mid x_t, D_t)$ provides a tractable lower-bound signal for the local future-information gain of $z_t$, up to the locality and coherence errors $\delta$ and $\epsilon$. This motivates using predictive entropy as the admission signal for the context.

**Redundancy.** When the context is full, the policy must remove a stored example. The ideal policy would delete the item that causes the smallest loss in future information:

$$u^* \in \arg\min_{u \in D_t} \left[ \mathcal{J}_t(D_t) - \mathcal{J}_t(D_t \setminus \{u\}) \right]. \tag{5}$$

This is impractical because it requires leave-one-out TFM evaluations for all stored examples. We therefore use redundancy as a proxy for low information loss. If two same-class examples are close in representation space, they are likely to provide overlapping evidence for future labels, so removing one of them should lose little information. This motivates removing close same-class examples, with the formal redundancy condition given in Assumption A.7.

# 4. CURE: Context Management for Streaming TFMs

Motivated by the three signals from the future-information view, we introduce CURE (Context management via Uncertainty-aware admission and Redundancy-aware Eviction), a context management policy for stream learning with TFMs. As depicted in Figure 1, CURE allocates part of the context budget to a short bank and the remaining budget to a long bank. New labeled examples first enter the short bank. When the short bank overflows, its oldest item becomes a candidate for long-term retention and is filtered by uncertainty-gated admission. When the long bank exceeds its budget, a redundant same-class example is removed.

## 4.1. Dual-Bank Context

The recency signal suggests that a context should preserve recent examples as a proxy for near-future query regions. To implement this, CURE maintains

$$D_t = S_t \cup L_t, \ |S_t| \leq B_S, \ |L_t| \leq B_L, \ B_S + B_L = B, \tag{6}$$

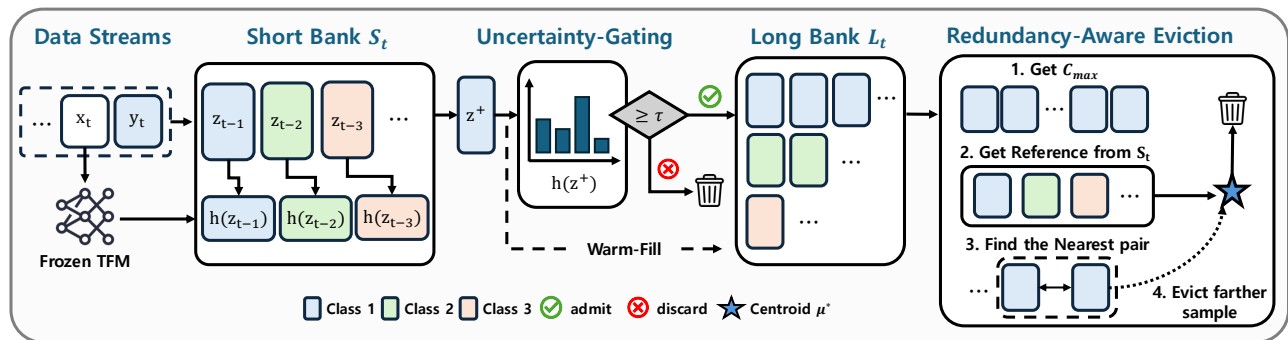

Figure 1. Overview of CURE. A new labeled example $z_t = (x_t, y_t)$ first enters the short bank $S_t$ to preserve recent support. When $S_t$ overflows, the oldest item $z^+$ becomes a long-bank candidate and is admitted according to its stored prediction-time entropy. When the long bank $L_t$ exceeds its budget, CURE removes a locally redundant same-class example.

where $S_t$ is a FIFO short bank and $L_t$ is a long bank. The short bank contains the most recent labeled examples as a sliding window, while the long bank stores older examples selected by uncertainty-gated admission and redundancy-aware eviction. Thus, every example remains available for at least the short-bank duration.

### 4.2. Entropy-Gated Long-Bank Admission

The uncertainty signal suggests that high-entropy examples can provide useful label information once revealed. We apply this signal to the long bank.

When an example $z_t = (x_t, y_t)$ is first predicted, the TFM outputs $p_t = q_\theta(\cdot \mid x_t, D_t)$ before observing $y_t$. We attach to $z_t$ its normalized predictive entropy

$$h(z_t) = \frac{-\sum_{c=1}^{C} p_t(c) \log p_t(c)}{\log C}, \qquad (7)$$

where $C$ is the number of classes. This score is computed once at prediction time and reused when the example later leaves the short bank.

Let $z^+ = (x^+, y^+)$ be the item that overflows from $S_t$. During *warm-fill*, candidates are inserted into $L_t$ until the long bank is full. Afterward, a candidate is admitted only if its stored entropy exceeds a threshold $\tau$:

$$\text{Admit}(z^+) = \begin{cases} 1, & |L_t| < B_L, \\ \mathbb{I}\{h(z^+) \geq \tau\}, & |L_t| \geq B_L. \end{cases} \qquad (8)$$

Thus, the long bank retains examples that were uncertain when first observed and are more likely to help future queries.

### 4.3. Redundancy-Aware Long-Bank Eviction

The redundancy signal suggests that close same-class examples provide overlapping evidence, so one can be removed with limited information loss. We again apply this signal only inside the long bank.

When $L_t$ exceeds its budget, CURE first selects the most represented class $c_{\max} \in \arg\max_c |\{z \in L_t : y_z = c\}|$, which reduces the risk of deleting sparse class evidence. Within this class, CURE finds the closest same-class pair in the normalized raw-feature representation $\phi(x)$:

$$(i^*, j^*) \in \arg \min_{\substack{i \neq j \\ z_i, z_j \in L_t \\ y_i = y_j = c_{\max}}} \|\phi(x_i) - \phi(x_j)\|_2. \qquad (9)$$

This pair is treated as the most redundant local evidence in the overrepresented class. To choose which endpoint to remove from $x_{i^*}$ and $x_{j^*}$, CURE uses the recent short bank as a reference for the current regime. It computes a recent centroid for class $c_{\max}$ and removes the endpoint farther from it. Full procedural details are provided in Appendix D.

## 5. Experiments

To evaluate whether CURE provides an effective context management protocol for streaming TFMs, we ask two questions. First, can bounded context management make TFMs competitive with classical stream learning methods? Second, does the policy transfer across different TFM backbones?

### 5.1. Experimental Setup

**Datasets.** We evaluate on seven streams: five real-world streams from the USP Data Stream repository[1] (NOAA, ME-TER, RIALTO, POSTURE-No8, POKER), one additional real-world stream (NOMAO), and one synthetic stream AGR(A) (Agrawal et al., 1993). POSTURE-No8 is a 10-class variant of POSTURE after removing the rarest class. Dataset statistics are provided in Appendix E.1.

**Models and baselines.** We use TabICL-v2 (Qu et al., 2026) as the primary backbone. We compare CURE against representative stream-learning baselines implemented in

---

[1] https://sites.google.com/view/uspdsrepository

*Table 1.* Prequential accuracy on data streams. Values in parentheses next to CURE indicate absolute gains over the best classical baseline and $\pm$ denotes standard deviation over five seeds. Best and second-best results are bolded and underlined.

| Dataset | CURE | ARF | BOLE | LevBag | SRP | EFDT | VFDT |
|---|---|---|---|---|---|---|---|
| NOAA | $\mathbf{81.94}_{(+2.51)}$ | 79.42±0.08 | 74.97±0.19 | 79.40±0.10 | 79.43±0.10 | 75.56 | 75.66 |
| METER | $\mathbf{90.80}_{(+19.31)}$ | 68.27±0.09 | 68.06±0.27 | 69.40±0.25 | 71.49±0.18 | 61.39 | 55.44 |
| RIALTO | $\mathbf{92.04}_{(+19.59)}$ | 63.40±0.12 | 53.62±0.32 | 64.13±0.18 | 72.45±0.15 | 58.73 | 40.46 |
| POSTURE-No8 | $\mathbf{62.10}_{(+1.58)}$ | 59.44±0.08 | 54.60±0.38 | 60.52±0.04 | 58.88±0.15 | 55.34 | 53.11 |
| POKER | $\mathbf{99.60}_{(+2.07)}$ | 84.05±0.11 | 88.46±0.19 | 97.53±0.10 | 89.29±0.16 | 82.75 | 89.09 |
| NOMAO | $\mathbf{97.87}_{(+0.45)}$ | 97.19±0.04 | 95.82±0.12 | 97.27±0.03 | 97.42±0.04 | 94.55 | 93.98 |
| AGR(A) | $\mathbf{90.93}_{(+0.67)}$ | 90.26±0.08 | 88.29±0.10 | 87.62±0.29 | 90.13±0.15 | 75.15 | 76.32 |
| Avg. rank | **1.00** | 3.71 | 5.43 | 3.14 | 2.57 | 6.00 | 6.14 |

*Table 2.* Backbone robustness across data streams. Values compare CURE with DualFIFO using the same TFM backbone. CURE improves in 17 of 18 comparisons.

| | LimiX-v1 | | TabPFN-v2.5 | | TabDPT-v1 | |
|---|---|---|---|---|---|---|
| Dataset | CURE | DualFIFO | CURE | DualFIFO | CURE | DualFIFO |
| NOAA | **81.43** | 80.83 | **81.82** | 81.26 | **81.94** | 81.49 |
| METER | **89.06** | 88.69 | **87.88** | 87.64 | **80.73** | 79.65 |
| RIALTO | **90.17** | 89.59 | **89.53** | 88.93 | **88.05** | 87.64 |
| POSTURE-No8 | **60.50** | 60.47 | 60.83 | **60.99** | **61.67** | 61.50 |
| NOMAO | **97.83** | 97.54 | **97.85** | 97.55 | **97.34** | 96.58 |
| AGR(A) | **90.56** | 90.54 | **91.02** | 90.85 | **90.02** | 89.93 |

MOA 24.07 (Bifet et al., 2010b): VFDT, EFDT, ARF, SRP, LevBag, and BOLE (Domingos & Hulten, 2000; Manapragada et al., 2018; Gomes et al., 2017; 2019; Bifet et al., 2010a; de Barros et al., 2016). To test transferability across TFMs, we also evaluate LimiX-v1, TabPFN-v2.5, and TabDPT-v1 (Zhang et al., 2025; Grinsztajn et al., 2025; Ma et al., 2024). Model and baseline details are given in Appendix E.2 and E.3.

**Evaluation protocol.** TFM policies use a total context budget of $B = 1000$, a short-bank ratio of $\rho = 0.75$, and a warm-up period of 100 stream steps. We update the context after every arriving labeled example and report cumulative prequential accuracy. Implementation details including all hyperparameters are provided in Appendix E.4.

### 5.2. Main Results

Table 1 compares CURE with TabICL-v2 against classical stream-learning baselines. CURE achieves the best prequential accuracy on all seven streams and obtains the best average rank. Its gains range from $+0.45$ points on NOMAO to $+19.31$ on METER and $+19.59$ on RIALTO. This corresponds to up to 27.0% improvement and 9.0% average relative improvement across datasets.

These results show that a TFM with bounded context management can be a strong stream learner compared with online tree and ensemble methods. This is notable because

CURE adapts to evolving streams only by updating the retained context.

### 5.3. Backbone Transferability

Table 2 evaluates whether the gains of CURE transfer across multiple TFM backbones. This matters because bounded context management operates at the input-context level and should not rely on a backbone-specific architecture.

Across LimiX-v1, TabPFN-v2.5, and TabDPT-v1, CURE improves over the prior method DualFIFO (Lourenço et al., 2026) in 17 of 18 comparisons. The gains are modest on AGR(A) and POSTURE-No8, but larger on NOAA, METER, RIALTO, and NOMAO. This indicates that the benefit is not specific to one backbone, and supports that CURE is a transferable interface for stream learning with TFMs.

### 5.4. Design-Space Ablation

We further compare CURE with controlled policy variants from the same design space that remove or alter uncertainty-gated admission and redundancy-aware eviction. CURE achieves the best average rank, suggesting that the proposed signals from the future-information view are complementary. Full definitions and results are provided in Appendix B.1.

## 6. Conclusion

This work studies stream learning with TFMs from the perspective of bounded context management. Unlike classical stream learners that adapt by updating model states, a TFM adapts through the labeled examples retained as context. We formalize this through a future-information view, which connects context usefulness to the information it provides about near-future queries. This leads to CURE, a simple policy that combines recent support, entropy-gated admission, and redundancy-aware eviction. Across multiple streams and TFM backbones, our results suggest that context management is a central mechanism for making TFMs effective on evolving data streams.

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

## A. Theory Details for the Future-Information View

This appendix provides the formal assumptions and proofs supporting Section 3. We fix a stream step $t$, a current context $D_t$, and a newly observed candidate $z_t = (x_t, y_t)$. The near-future feature distribution is

$$\mathcal{P}_{t,X}^+ = \frac{1}{h} \sum_{s \in \mathcal{H}_t^+} \mathcal{P}_{s,X}, \qquad \mathcal{H}_t^+ = \{t+1, \ldots, t+h\}.$$

For a future feature value $x'$, let $Y_{x'}$ denote its label random variable. For the current candidate feature $x_t$, we use $Y_{x_t}$ to denote the label random variable before its true label is observed.

### A.1. Recent-Window Approximation

The recency signal is motivated by replacing the unavailable near-future feature distribution with the empirical distribution of recent examples.

Let $S_t$ be the recent window of labeled examples before time $t$, and let $\widehat{\mathcal{P}}_{S_t}$ be its empirical feature distribution. Replacing the unknown near-future feature distribution with the recent window gives the recent-window objective

$$\widehat{\mathcal{J}}_{S_t}(D_t) = \mathbb{E}_{x_t \sim \widehat{\mathcal{P}}_{S_t}} \left[ I(D_t; Y_{x_t} \mid x_t) \right]. \tag{10}$$

The following assumption states the condition under which the recent window can act as a proxy for the near future: the two feature distributions should not be too far apart.

**Assumption A.1** (Recent-window local stability). At time $t$, the empirical feature distribution of the recent window is close to the near-future feature distribution:

$$W_1 \left( \widehat{\mathcal{P}}_{S_t}, \mathcal{P}_{t,X}^+ \right) \leq \rho_{S_t}, \tag{11}$$

where $W_1$ is the Wasserstein-1 distance in the representation space and $\rho_{S_t}$ is the discrepancy between the recent-window and near-future feature distributions at time $t$.

Under this condition, the following lemma shows that evaluating future information on the recent window is close to evaluating it on the ideal near-future distribution.

**Lemma A.2** (Recent-window approximation). *For a fixed context $D_t$, suppose the query-wise information function $x \mapsto I(D_t; Y_x \mid x)$ is $L$-Lipschitz in the representation space. Under Assumption A.1,*

$$\left| \mathcal{J}_t(D_t) - \widehat{\mathcal{J}}_{S_t}(D_t) \right| \leq L\rho_{S_t}. \tag{12}$$

This bound is the formal reason for reserving a short-bank budget: if the recent window is close to the near future, the information measured on recent examples is a reliable proxy for near-future information.

*Proof.* Define the query-wise information function for the current context as

$$f_{D_t}(x) = I(D_t; Y_x \mid x).$$

By assumption, $f_{D_t}$ is $L$-Lipschitz in the representation space. The near-future objective and the recent-window objective can be written as

$$\mathcal{J}_t(D_t) = \mathbb{E}_{x \sim \mathcal{P}_{t,X}^+}[f_{D_t}(x)], \qquad \widehat{\mathcal{J}}_{S_t}(D_t) = \mathbb{E}_{x \sim \widehat{\mathcal{P}}_{S_t}}[f_{D_t}(x)].$$

By the Kantorovich–Rubinstein duality for Wasserstein-1 distance,

$$|\mathbb{E}_{x \sim P}[f_{D_t}(x)] - \mathbb{E}_{x \sim Q}[f_{D_t}(x)]| \leq L W_1(P, Q)$$

for any two feature distributions $P$ and $Q$. Taking $P = \mathcal{P}_{t,X}^+$ and $Q = \widehat{\mathcal{P}}_{S_t}$ gives

$$\left| \mathcal{J}_t(D_t) - \widehat{\mathcal{J}}_{S_t}(D_t) \right| \leq L W_1(\widehat{\mathcal{P}}_{S_t}, \mathcal{P}_{t,X}^+).$$

Applying Assumption A.1 yields Eq. (12). $\qquad\square$

## A.2. Marginal Future Information

The context-level objective $\mathcal{J}_t(D_t)$ scores an entire context, but the online policy must decide whether a single new example should be kept. The next lemma connects this instance-level decision to future-label information.

For the newly observed candidate $z_t = (x_t, y_t)$, define its marginal future information by

$$\Delta_t(z_t \mid D_t) = \mathcal{J}_t(D_t \cup \{z_t\}) - \mathcal{J}_t(D_t).$$

**Lemma A.3** (Item contribution as future-label information). *For the candidate $z_t = (x_t, y_t)$,*

$$\Delta_t(z_t \mid D_t) = \mathbb{E}_{x' \sim \mathcal{P}^+_{t,X}} \left[ I(Y_{x_t}; Y_{x'} \mid x_t, x', D_t) \right]. \tag{13}$$

This identity says that a candidate is useful when its label carries information about labels of near-future queries. It is the bridge from the context-level objective to an admission rule for individual examples.

*Proof.* By definition,

$$\begin{aligned}
\Delta_t(z_t \mid D_t) &= \mathcal{J}_t(D_t \cup \{z_t\}) - \mathcal{J}_t(D_t) \\
&= \mathbb{E}_{x' \sim \mathcal{P}^+_{t,X}} \left[ I(D_t \cup \{z_t\}; Y_{x'} \mid x') - I(D_t; Y_{x'} \mid x') \right].
\end{aligned}$$

The candidate $z_t$ is represented by its feature value $x_t$ and label random variable $Y_{x_t}$. By the chain rule for mutual information,

$$I(D_t \cup \{z_t\}; Y_{x'} \mid x') = I(D_t; Y_{x'} \mid x') + I(Y_{x_t}; Y_{x'} \mid x_t, x', D_t).$$

Substituting this identity into the previous expression gives Eq. (13). $\square$

## A.3. Entropy Lower Bound for Local Information

Lemma A.3 gives an ideal admission criterion, but it still depends on unknown future labels. We therefore relate this quantity to a tractable prediction-time signal: the entropy of the candidate before its label is observed.

Let $\mathcal{N}_t(z_t) \subseteq \mathcal{X}$ denote a measurable local region around $x_t$. We call it the effective future region of $z_t$, where the revealed label of $z_t$ can provide local evidence for near-future queries. Define its near-future mass as

$$\alpha_t(z_t) = \Pr_{x' \sim \mathcal{P}^+_{t,X}} \left[ x' \in \mathcal{N}_t(z_t) \right]. \tag{14}$$

We first require the candidate's prediction-time uncertainty to be representative of the uncertainty in its local future region.

**Assumption A.4** (Local entropy consistency). For the newly observed candidate $z_t = (x_t, y_t)$,

$$\mathbb{E}\left[ H(Y_{x'} \mid x_t, x', D_t) \mid x' \in \mathcal{N}_t(z_t) \right] \geq H(Y_{x_t} \mid x_t, D_t) - \delta. \tag{15}$$

This condition says that the candidate should not be much more uncertain than the nearby future queries it is intended to support. The slack variable $\delta$ captures violations of this local entropy consistency.

We also require the revealed label of the candidate to be informative about labels in its local future region.

**Assumption A.5** (Local label coherence). For the newly observed candidate $z_t = (x_t, y_t)$,

$$\mathbb{E}\left[ H(Y_{x'} \mid Y_{x_t}, x_t, x', D_t) \mid x' \in \mathcal{N}_t(z_t) \right] \leq \epsilon. \tag{16}$$

This assumption does not require labels to be deterministic functions of the input. Rather, the residual term $\epsilon$ captures stochastic labels, unobserved variables, and class overlap within the local region. When nearby labels are noisy or only weakly coherent, $\epsilon$ becomes larger and the resulting lower bound becomes weaker.

Together, these two local conditions connect prediction-time entropy to the future-label information supplied by $z_t$.

**Theorem A.6** (Entropy lower bound for local information). *For a candidate $z_t = (x_t, y_t)$ and its effective future region $\mathcal{N}_t(z_t)$, the expected local information gain satisfies*

$$
\mathbb{E}_{x' \sim \mathcal{P}_{t,X}^+} \left[ \mathbf{1}\{x' \in \mathcal{N}_t(z_t)\} I(Y_{x_t}; Y_{x'} \mid x_t, x', D_t) \right]
$$
$$
\geq \alpha_t(z_t) \left[ H(Y_{x_t} \mid x_t, D_t) - \delta - \epsilon \right]. \tag{17}
$$

Theorem A.6 provides the formal motivation for entropy-gated admission. It shows that, when local entropy consistency and local label coherence approximately hold, higher prediction-time entropy yields a larger lower-bound signal for the local future-information gain of $z_t$.

*Proof.* For any $x' \in \mathcal{N}_t(z_t)$, the definition of conditional mutual information gives

$$
I(Y_{x_t}; Y_{x'} \mid x_t, x', D_t) = H(Y_{x'} \mid x_t, x', D_t) - H(Y_{x'} \mid Y_{x_t}, x_t, x', D_t).
$$

Taking expectation over near-future feature values and restricting to the effective future region,

$$
\mathbb{E}_{x' \sim \mathcal{P}_{t,X}^+} \left[ \mathbf{1}\{x' \in \mathcal{N}_t(z_t)\} I(Y_{x_t}; Y_{x'} \mid x_t, x', D_t) \right]
$$
$$
= \alpha_t(z_t) \mathbb{E} \left[ I(Y_{x_t}; Y_{x'} \mid x_t, x', D_t) \mid x' \in \mathcal{N}_t(z_t) \right]
$$
$$
= \alpha_t(z_t) \mathbb{E} \left[ H(Y_{x'} \mid x_t, x', D_t) - H(Y_{x'} \mid Y_{x_t}, x_t, x', D_t) \mid x' \in \mathcal{N}_t(z_t) \right].
$$

By Assumption A.4,
$$
\mathbb{E} \left[ H(Y_{x'} \mid x_t, x', D_t) \mid x' \in \mathcal{N}_t(z_t) \right] \geq H(Y_{x_t} \mid x_t, D_t) - \delta.
$$

By Assumption A.5,
$$
\mathbb{E} \left[ H(Y_{x'} \mid Y_{x_t}, x_t, x', D_t) \mid x' \in \mathcal{N}_t(z_t) \right] \leq \epsilon.
$$

Combining these two inequalities proves Eq. (17). □

### A.4. Same-Class Evidence Redundancy

The removal problem asks which stored example can be deleted with minimal loss in future information. The following assumption formalizes the intuition that close same-class examples provide overlapping evidence.

**Assumption A.7** (Same-class evidence redundancy). For two same-class context items $z_i = (x_i, y_i)$ and $z_j = (x_j, y_j)$ in $D_t$ with $y_i = y_j$, there exists a nondecreasing function $\eta$ with $\eta(r) \to 0$ as $r \to 0$ such that

$$
\mathcal{J}_t(D_t) - \mathcal{J}_t(D_t \setminus \{z_j\}) \leq \eta(\|\phi(x_i) - \phi(x_j)\|_2). \tag{18}
$$

This assumption says that if a stored example has a close same-class neighbor in representation space, then removing it causes only a small loss of future information. This is the formal motivation for removing close same-class examples from the long bank when the budget is full.

## B. Additional Analyses

### B.1. Controlled Policy Variants

We first evaluate controlled context-policy variants to isolate the design choices of CURE. The variants summarized in Table 3 test whether uncertainty is needed for retaining informative candidates, whether redundancy is needed for low-loss removal, and whether entropy and same-class nearest-neighbor removal are the right instantiations of these signals. All variants use the same TFM backbone and the same total context budget. We exclude POKER from this design-space evaluation due to its large stream length and substantially longer runtime.

**DualFIFO** is the FIFO-style dual-memory reference policy. It uses the same short-bank and long-bank budgets as CURE. New examples enter the short bank, and short-bank overflow candidates are inserted into the long bank. When the long bank exceeds its budget, DualFIFO selects the most represented class and removes the oldest long-bank item from that class. Thus, DualFIFO preserves recency and coarse class balance, but does not use uncertainty or redundancy-aware removal.

*Table 3.* TFM context-policy variants for the design-space evaluation. They differ only in how they instantiate uncertainty-based retention and redundancy-aware removal.

| Policy | Recency | Long bank | Retention signal | Removal rule |
|---|---|---|---|---|
| CURE | ✓ | ✓ | entropy | same-class NN |
| DualFIFO | ✓ | ✓ | – | class-aware FIFO |
| Entropy-only | ✓ | ✓ | entropy | class-aware FIFO |
| Redundancy-only | ✓ | ✓ | – | same-class NN |
| CURE-Margin | ✓ | ✓ | margin | same-class NN |
| CURE-GlobalNN | ✓ | ✓ | entropy | global NN |

*Table 4.* Design-space ablation with prequential accuracy. Best results are bolded and second-best results are underlined. CURE achieves the best average rank, showing that the theory-driven signals are more effective when used together than in isolation.

| Dataset | CURE | DualFIFO | Entropy-only | Redundancy-only | CURE-Margin | CURE-GlobalNN |
|---|---|---|---|---|---|---|
| NOAA | 81.94 | 81.42 | 81.78 | 81.64 | 81.90 | **81.95** |
| METER | **90.80** | 90.60 | 89.94 | 90.63 | 90.53 | 90.68 |
| RIALTO | **92.04** | 91.72 | 91.36 | 91.97 | 92.00 | 92.02 |
| POSTURE-No8 | **62.10** | 61.85 | 61.97 | 62.01 | 62.01 | 61.88 |
| NOMAO | **97.87** | 97.47 | 97.65 | 97.64 | 97.81 | 97.75 |
| AGR(A) | **90.93** | 90.45 | 90.84 | 90.54 | 90.74 | 90.62 |
| Avg. rank | **1.17** | 5.50 | 4.33 | 4.08 | 3.08 | 2.83 |

**Entropy-only** isolates uncertainty-based retention. It uses the same entropy-gated long-bank admission rule as CURE, but keeps the DualFIFO class-aware FIFO removal rule when the long bank overflows. This tests whether uncertainty-based candidate filtering alone is sufficient without redundancy-aware capacity management.

**Redundancy-only** isolates redundancy-aware removal. It admits every short-bank overflow candidate into the long bank, but uses the same-class nearest-neighbor removal rule from CURE when the long bank overflows. This tests whether removing duplicated same-class evidence is sufficient without uncertainty-based candidate filtering.

**CURE-Margin** tests the choice of uncertainty score. It keeps the same dual-memory structure and same-class redundancy-aware removal rule as CURE, but replaces predictive entropy with a top-two margin score. For a prediction distribution $p_t = q_\theta(\cdot \mid x_t, D_t)$, let $p_{t,(1)}$ and $p_{t,(2)}$ be the largest and second-largest class probabilities. The margin-based uncertainty score is

$$h_{\mathrm{margin}}(z_t) = 1 - \big(p_{t,(1)} - p_{t,(2)}\big).$$

A larger value indicates a smaller gap between the two most likely classes.

**CURE-GlobalNN** tests the label-conditional nature of redundancy. It keeps entropy-gated admission but replaces same-class nearest-neighbor removal with class-agnostic nearest-neighbor removal. When the long bank overflows, it finds

$$(i^*, j^*) \in \arg \min_{\substack{i \neq j \\ z_i, z_j \in L_t}} \|\phi(x_i) - \phi(x_j)\|_2.$$

It then removes one endpoint using the same centroid-based tie-breaking rule as CURE. This variant tests whether redundancy should be defined purely geometrically or conditioned on the label.

Table 4 compares CURE with these controlled policy variants on six streams. CURE improves over DualFIFO on all six streams, achieves the best result on five, and has the best average rank. Because DualFIFO already uses the same dual-memory structure with recency and class-coverage bias, these gains indicate the efficacy of the future-information-guided design beyond FIFO retention.

The component variants show that uncertainty and redundancy are complementary. Entropy-only helps on some streams, but can be unstable when removal remains age-based. Redundancy-only consistently improves over DualFIFO, but does not reach CURE. The design-choice variants further support our specific instantiation since predictive entropy generally performs better than the top-two margin score, and same-class nearest-neighbor removal gives a better average rank than class-agnostic removal.

*Table 5.* Sensitivity of CURE to the entropy threshold $\tau$ with TabICLv2. Values are prequential accuracies.

| Dataset | $\tau = 0$ | $\tau = 0.1$ | $\tau = 0.2$ | $\tau = 0.3$ | $\tau = 0.4$ | $\tau = 0.5$ |
|---|---|---|---|---|---|---|
| NOAA | 81.64 | 81.64 | 81.83 | 81.82 | **81.94** | 81.88 |
| METER | 90.63 | **90.80** | 90.62 | 90.68 | 90.62 | 90.48 |
| RIALTO | 91.97 | **92.04** | 91.98 | 92.00 | 92.02 | 91.96 |
| POSTURE-No8 | 62.01 | 62.03 | **62.10** | 62.03 | 62.03 | 62.08 |
| NOMAO | 97.64 | 97.71 | 97.82 | **97.87** | 97.86 | 97.86 |
| AGR(A) | 90.54 | 90.49 | 90.61 | 90.52 | **90.93** | 90.61 |

### B.2. Sensitivity to the Entropy Threshold $\tau$

Table 5 studies the sensitivity of CURE to the entropy threshold $\tau$. The $\tau = 0$ column disables uncertainty-gated admission and therefore corresponds to the Redundancy-only variant. Moving from $\tau = 0$ to a positive threshold improves the best accuracy on all streams, indicating that the uncertainty gate contributes beyond redundancy-aware eviction alone.

At the same time, CURE is not sensitive to a narrowly tuned threshold. Across positive thresholds $\tau \in \{0.1, 0.2, 0.3, 0.4, 0.5\}$, the largest within-dataset accuracy difference is only 0.39 percentage points. Thus, uncertainty-gated admission is useful, while the method remains stable over a broad range of positive thresholds.

*Table 6.* Average per-step runtime over seven streams. Values are in seconds.

| Method | Total | Fit | Predict | Evict |
|---|---|---|---|---|
| CURE | 0.0283 | 0.0044 | 0.0226 | 0.0013 |
| DualFIFO | 0.0259 | 0.0040 | 0.0214 | 0.0005 |

### B.3. Efficiency Analysis

Table 6 reports the average per-step runtime of CURE and DualFIFO, decomposed into context fitting, prediction, and eviction. The reported "Total" column sums the measured core components used by the streaming prediction loop.

CURE introduces additional computation over DualFIFO because it performs entropy-gated long-bank admission and redundancy-aware eviction, whereas DualFIFO updates the long bank using FIFO-style rules. Nevertheless, the absolute overhead is small. The average total step time increases from 0.0259 seconds to 0.0283 seconds, an increase of about 0.0024 seconds per example. The eviction component itself takes only 0.0013 seconds per step on average for CURE, while prediction takes 0.0226 seconds. Thus, most of the runtime remains dominated by the frozen TFM context-fitting and prediction calls, not by the redundancy-aware removal logic. This suggests that the additional policy logic in CURE is practical for the evaluated streaming setting.

## C. Related Work

**Classical stream learning.** Supervised stream learning is commonly studied under the test-then-train prequential protocol, where each example must be predicted before its label is observed. Classical baselines address this setting by updating the learner state. Hoeffding-tree methods such as VFDT and EFDT maintain node-level sufficient statistics and use statistical tests to decide when to grow or revise tree splits (Domingos & Hulten, 2000; Manapragada et al., 2018). Ensemble methods such as Leveraging Bagging, BOLE, Adaptive Random Forests, and Streaming Random Patches maintain multiple online learners and adapt them through resampling, randomization, and drift-aware replacement mechanisms (Bifet et al., 2010a; de Barros et al., 2016; Gomes et al., 2017; 2019). Recent stream-learning work further studies dynamic ensemble diversification, online concept-drift detection, and neural stream classifiers (Abadifard et al., 2023; Wan et al., 2024; Su et al., 2024). These approaches adapt by changing model states, whereas our work studies adaptation through the retained context of a TFM.

**Tabular foundation models.** Tabular foundation models are in-context predictors that adapt to new datasets by conditioning on labeled examples, rather than by performing dataset-specific training. This paradigm follows prior-data fitted networks, where a transformer is pretrained to approximate Bayesian posterior predictive inference from context examples (Müller et al., 2021). TabPFN introduced this idea for tabular classification using synthetic prior-generated

tables (Hollmann et al., 2022), and recent TFMs such as TabICL, LimiX, and TabDPT extend the paradigm with improved architectures, training procedures, or pretraining data (Qu et al., 2025; Zhang et al., 2025; Ma et al., 2024). Because TFMs expose the labeled context as their adaptation interface, they are naturally suited to bounded-memory stream learning.

## D. Algorithmic Details

This appendix gives the procedural form of the context update used by CURE. The short bank $S$ implements the recency signal, while the long bank $L$ stores older examples selected by entropy-gated admission and redundancy-aware eviction. The stored score $h(z)$ is the prediction-time entropy of $z$, computed when $z$ was first observed and later used when the sample overflows from the short bank.

---

**Algorithm 1** CURE memory update

---

**Require:** Short bank $S$, long bank $L$, new labeled sample $z = (x, y)$, entropy $h(z)$, capacities $B_S, B_L$, threshold $\tau$
**Ensure:** Updated short and long banks $S, L$

1:   $S \leftarrow S \cup \{z\}$
2:   **if** $|S| > B_S$ **then**
3:      $z^+ = (x^+, y^+) \leftarrow$ oldest sample in $S$
4:      $S \leftarrow S \setminus \{z^+\}$
5:      **if** $|L| < B_L$ **or** $h(z^+) \geq \tau$ **then**
6:         $L \leftarrow L \cup \{z^+\}$
7:      **end if**
8:   **end if**
9:   **if** $|L| > B_L$ **then**
10:     $c_{\max} \leftarrow \arg\max_c |\{(x_i, y_i) \in L : y_i = c\}|$
11:     $(i^*, j^*) \leftarrow \arg\min_{\substack{i \neq j \\ z_i, z_j \in L \\ y_i = y_j = c_{\max}}} \|\phi(x_i) - \phi(x_j)\|_2$
12:     $\mu^* \leftarrow \begin{cases} |S(c_{\max})|^{-1} \sum_{z_i \in S(c_{\max})} \phi(x_i), & |S(c_{\max})| > 0, \\ |S|^{-1} \sum_{z_i \in S} \phi(x_i), & |S(c_{\max})| = 0 \end{cases}$
13:     $u^* \leftarrow \arg\max_{u \in \{i^*, j^*\}} \|\phi(x_u) - \mu^*\|_2$
14:     $L \leftarrow L \setminus \{z_{u^*}\}$
15:   **end if**
16:   **return** $S, L$

---

The nearest-pair search implements the redundancy signal: among the overrepresented class in the long bank, the closest same-class pair is treated as the most redundant evidence, and the item farther from the recent reference centroid is removed.

## E. Experimental Details

This appendix provides additional details on datasets, TFM backbones, classical stream-learning baselines, context-management policy variants, and evaluation settings.

### E.1. Datasets

We evaluate seven datasets in total with six real-world data streams and one synthetic abrupt-drift stream. For real-world streams, we preserve the original row order and do not shuffle examples. Each file is loaded as a headerless CSV file, then the last column is treated as the class label while all preceding columns are used as input features. The METER and POSTURE loaders drop one malformed row before constructing the stream. A synthetic stream AGR(A) is generated from the Agrawal stream generator. We generated 30,000 examples with 9 features and 2 classes. The concept function changes abruptly at steps 7,500, 15,000, and 22,500 following the sequence $(0, 3, 6, 9)$, with $10\%$ feature perturbation.

Table 7 summarizes the datasets used in our experiments. We categorize class balance using the imbalance ratio

$$\mathrm{IR} = \frac{\max_c n_c}{\min_c n_c}.$$

*Table 7.* Dataset statistics after applying the current stream loader.

| Dataset | Rows | Features | Classes | Balance |
|---|---|---|---|---|
| NOAA | 18,159 | 8 | 2 | imbalanced |
| METER | 22,948 | 96 | 10 | balanced |
| RIALTO | 82,250 | 27 | 10 | balanced |
| POSTURE-No8 | 163,477 | 3 | 10 | highly imbalanced |
| POKER | 829,201 | 10 | 10 | extremely imbalanced |
| NOMAO | 34,465 | 118 | 2 | imbalanced |
| AGR(A) | 30,000 | 9 | 2 | synthetic abrupt drift |

A dataset is categorized as balanced if $IR \leq 1.1$, imbalanced if $1.1 < IR \leq 10$, highly imbalanced if $10 < IR \leq 100$, and extremely imbalanced if $IR > 100$.

### E.2. Backbones

We evaluate CURE with four frozen tabular foundation model backbones: TabICL-v2, TabPFN-v2.5, TabDPT-v1, and LimiX-v1 to test the backbone-agnostic properties of CURE.

**TabICL-v2.** TabICL was designed to scale tabular in-context learning beyond the expensive alternating row/column attention, using a two-stage column-then-row attention module to build row embeddings before efficient in-context prediction (Qu et al., 2025). TabICL-v2 further improves scalability and performance with a more diverse synthetic data engine and optimized pretraining protocols for both classification and regression (Qu et al., 2026). We use TabICL-v2 as the primary backbone in the main experiments.

**TabPFN-v2.5.** TabPFN-v1 instantiated Prior-Data Fitted Networks for tabular classification. TabPFN-v2 introduced a stronger synthetic task distribution and an alternating-attention architecture over samples and features (Hollmann et al., 2025). TabPFN-v2.5 follows this line and scales it toward larger tabular contexts (Grinsztajn et al., 2025).

**TabDPT-v1.** TabDPT targets the limitation of purely synthetic pretraining by training tabular in-context learning architectures on real data with self-supervised learning and retrieval (Ma et al., 2024). We use TabDPT-v1 as the real-data-pretrained backbone.

**LimiX-v1.** LimiX treats structured data as a joint distribution over variables and missingness, enabling classification, regression, imputation, and generation through query-based conditional prediction (Zhang et al., 2025). It is pretrained with masked joint-distribution modeling under an episodic context-conditional objective. We use the LimiX-16M checkpoint and the default no-retrieval classification configuration.

### E.3. Baselines

We compare CURE against various classical online stream-learning baselines. These baselines are implemented using MOA 24.07.

**VFDT.** The Very Fast Decision Tree (VFDT), also known as the Hoeffding Tree, is a milestone incremental decision-tree algorithm for high-speed data streams. It uses Hoeffding bounds to decide when the best split attribute is statistically reliable, enabling decision-tree induction with constant memory and constant time per example in the idealized setting (Domingos & Hulten, 2000).

**EFDT.** The Extremely Fast Decision Tree (EFDT), or Hoeffding Anytime Tree, modifies Hoeffding Tree by allowing splits to be made earlier and later revised when better split choices become statistically supported. Compared with the conservative split policy of VFDT, EFDT is more statistically efficient and often obtains stronger prequential accuracy at modest additional computational cost (Manapragada et al., 2018).

*Table 8.* Selected hyperparameters for the main comparison. For MOA baselines, $s$ is ensemble size, $g$ is grace period, and $t$ is tie threshold.

| Dataset | CURE $\tau$ | ARF | BOLE | LevBag | SRP | EFDT | VFDT |
|---|---|---|---|---|---|---|---|
| NOAA | 0.4 | $s90, g400, t0.1$ | $s90, g100, t0.1$ | $s90, g100, t0.1$ | $s90, g400, t0.1$ | $g400, t0.1$ | $g400, t0.1$ |
| METER | 0.1 | $s90, g400, t0.1$ | $s90, g1000, t0.1$ | $s90, g100, t0.1$ | $s90, g100, t0.1$ | $g1000, t0.01$ | $g400, t0.1$ |
| RIALTO | 0.1 | $s90, g100, t0.1$ | $s90, g100, t0.1$ | $s90, g100, t0.1$ | $s90, g100, t0.1$ | $g100, t0.1$ | $g400, t0.1$ |
| POSTURE-No8 | 0.2 | $s90, g100, t0.1$ | $s90, g100, t0.1$ | $s90, g100, t0.1$ | $s90, g100, t0.1$ | $g400, t0.1$ | $g100, t0.1$ |
| POKER | 0.3 | $s90, g100, t0.1$ | $s90, g100, t0.1$ | $s90, g100, t0.1$ | $s90, g100, t0.1$ | $g100, t0.1$ | $g400, t0.1$ |
| NOMAO | 0.3 | $s90, g100, t0.1$ | $s90, g1000, t0.01$ | $s90, g400, t0.1$ | $s90, g100, t0.1$ | $g100, t0.05$ | $g100, t0.1$ |
| AGR(A) | 0.4 | $s90, g100, t0.1$ | $s90, g100, t0.1$ | $s90, g100, t0.1$ | $s90, g100, t0.01$ | $g100, t0.1$ | $g400, t0.1$ |

**ARF.** Adaptive Random Forest (ARF) is a strong ensemble baseline for evolving data stream classification. It combines online resampling with multiple incremental tree learners and uses adaptive operators to replace underperforming trees under concept drift (Gomes et al., 2017).

**SRP.** Streaming Random Patches (SRP) is an ensemble method designed for evolving streams that combines online bagging with random subspaces. Unlike methods that only randomize samples or only randomize features, SRP can jointly exploit instance resampling and feature-subspace diversity (Gomes et al., 2019).

**Leveraging Bagging.** Leveraging Bagging extends online bagging with stronger randomization to increase ensemble diversity. It was proposed for evolving data streams as a simple but effective bagging variant with additional randomness and drift-aware mechanisms (Bifet et al., 2010a).

**BOLE.** The Boosting-like Online Learning Ensemble (BOLE) adapts boosting-style ideas to concept-drifting data streams. It modifies online boosting mechanisms to better handle changing distributions and maintain ensemble performance under drift (de Barros et al., 2016).

### E.4. Evaluation Details

**Protocol.** All methods are evaluated under prequential protocol. At step $t$, the learner predicts the label of $x_t$ using its current model or context, then receives the true label $y_t$, and finally updates its model or context using the labeled example $z_t = (x_t, y_t)$. We use a warm-up period of 100 stream steps and report prequential accuracy only after warm-up.

**Hardware.** All TFM-based streaming experiments are run on a single NVIDIA H200 GPU.

**MOA baseline hyperparameter grid.** For all MOA baseline runs, we fix the split confidence to $10^{-7}$ and use `NBAdaptive` leaf prediction.

For ensemble baselines, ARF, BOLE, LevBag, and SRP, we fix the ensemble size to 90 and grid search over grace period and tie threshold:

$$g \in \{100, 400, 1000\}, \qquad t \in \{0.01, 0.05, 0.1\}.$$

Thus, each ensemble baseline is evaluated with $3 \times 3 = 9$ configurations per dataset. For tree baselines, EFDT and VFDT, we use the same grid over grace period and tie threshold:

$$g \in \{100, 400, 1000\}, \qquad t \in \{0.01, 0.05, 0.1\}.$$

This also gives 9 configurations per tree baseline.

**Selected Hyperparameters** Table 8 reports the hyperparameters selected for the main results in Table 1. For CURE, the selected value is the entropy threshold $\tau$. For the MOA baselines, $s$ denotes the ensemble size, $g$ denotes the grace period, and $t$ denotes the tie threshold. The ensemble size is fixed to $s = 90$ for ARF, BOLE, LevBag, and SRP. EFDT and VFDT are single-tree methods and therefore do not use $s$.

# F. Additional Visualizations

### F.1. Prequential Accuracy Trajectories

Figures 2–8 show prequential accuracy trajectories of CURE and the MOA baselines and verify that CURE's advantage is persistent over time. On METER, RIALTO, POSTURE-No8, and POKER, CURE quickly separates from the classical baselines and maintains a large gap throughout most of the stream. On NOAA, NOMAO, and AGR(A), where the best classical baselines are closer, CURE still remains among the top trajectories and avoids the stronger degradation observed for some tree and ensemble methods. This supports the interpretation that the gains in Table 1 reflect stable stream-level behavior rather than a final-score artifact.

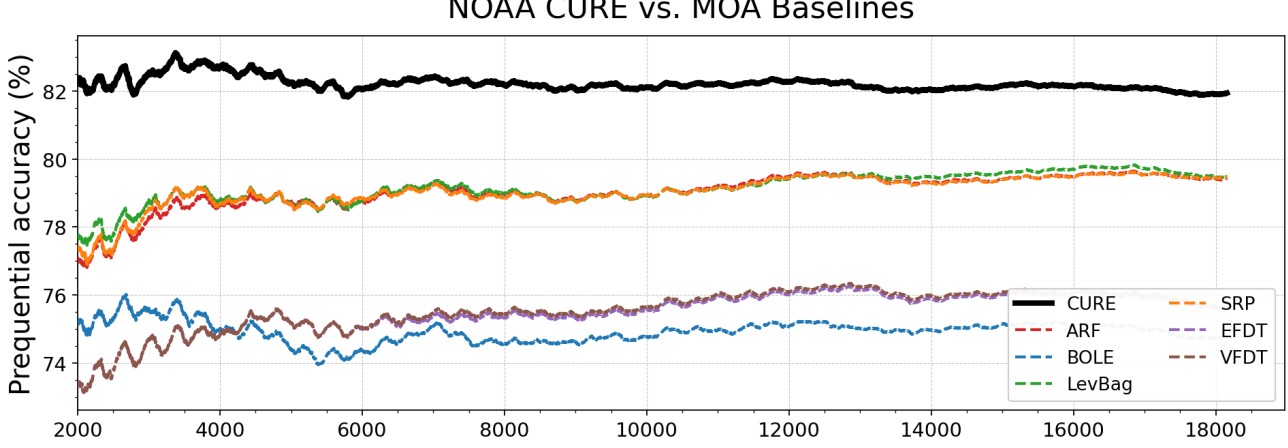

*Figure 2.* Prequential accuracy trajectories of CURE and MOA baselines on NOAA.

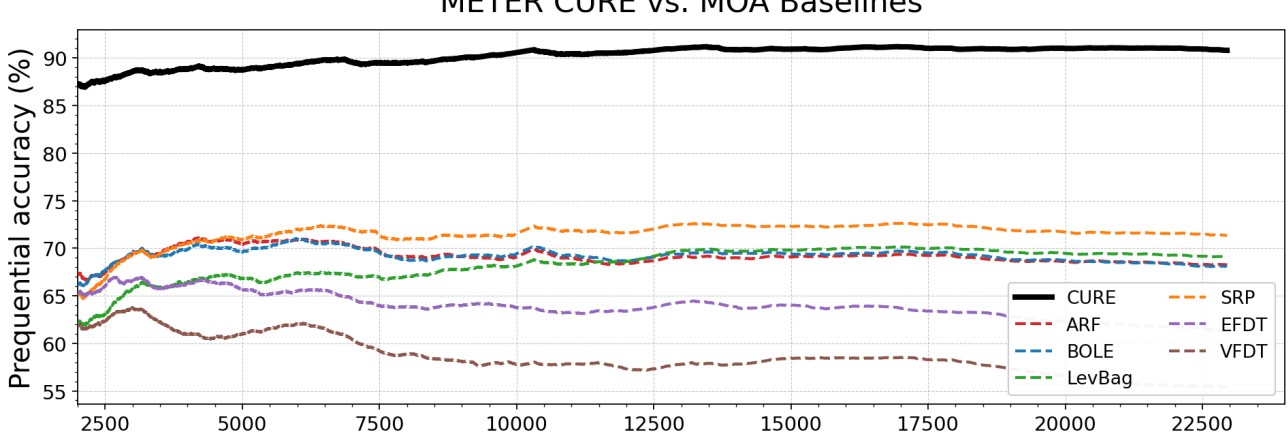

*Figure 3.* Prequential accuracy trajectories of CURE and MOA baselines on METER.

## RIALTO CURE vs. MOA Baselines

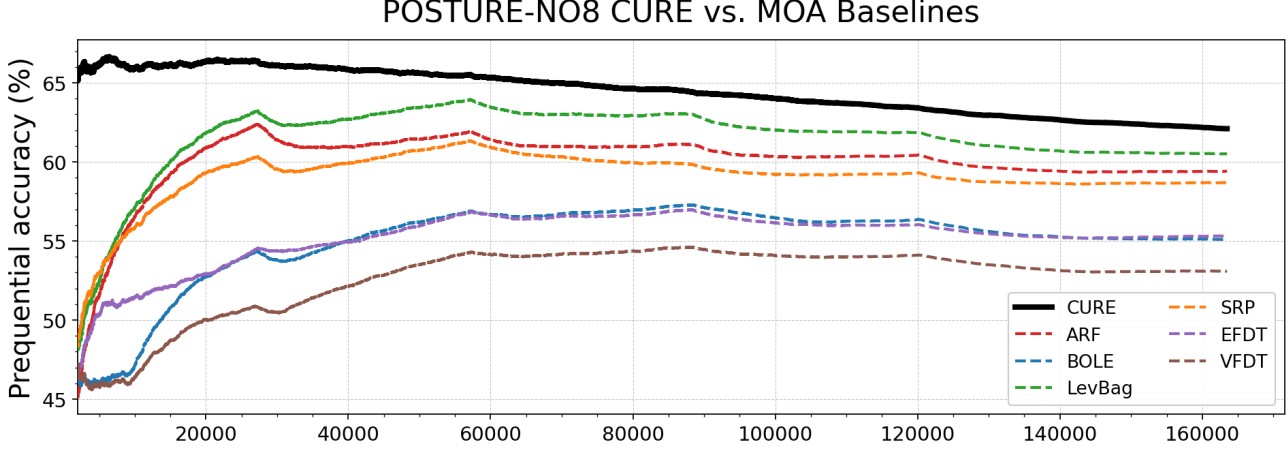

*Figure 4.* Prequential accuracy trajectories of CURE and MOA baselines on RIALTO.

## POSTURE-NO8 CURE vs. MOA Baselines

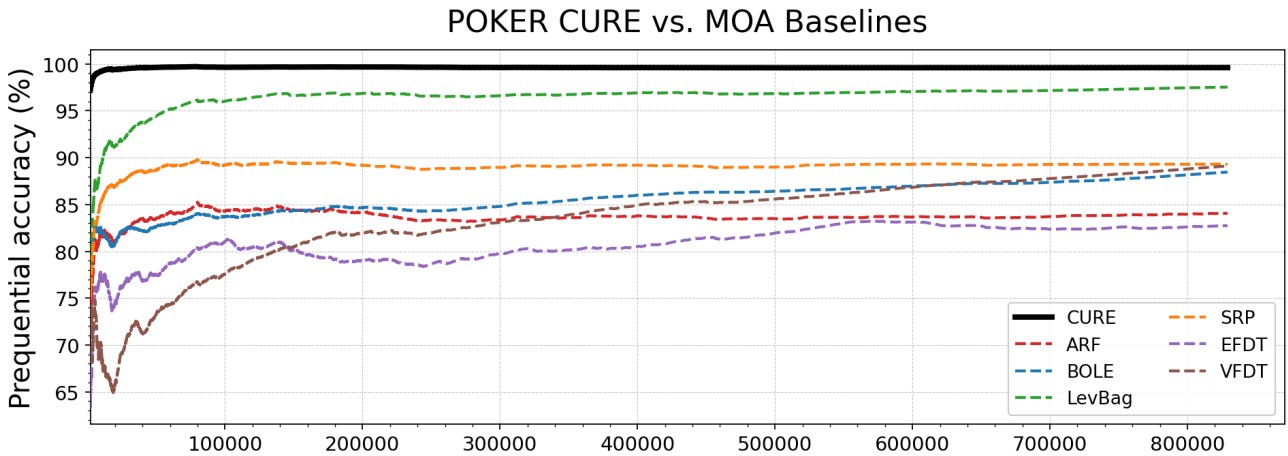

*Figure 5.* Prequential accuracy trajectories of CURE and MOA baselines on POSTURE-No8.

## POKER CURE vs. MOA Baselines

*Figure 6.* Prequential accuracy trajectories of CURE and MOA baselines on POKER.

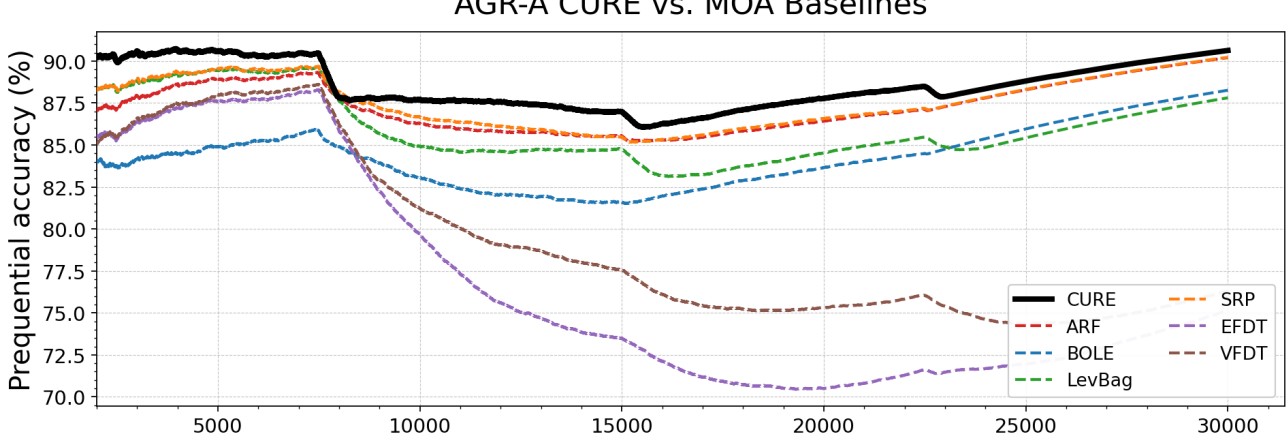

*Figure 7.* Prequential accuracy trajectories of CURE and MOA baselines on NOMAO.

*Figure 8.* Prequential accuracy trajectories of CURE and MOA baselines on AGR(A).

