# OpenReview forum: "Bounded Context Management for Tabular Foundation Models on Stream Learning"
_ICML.cc/2026/Workshop/FMSD — FMSD @ ICML 2026 SpotlightOral_

### Official Review · Reviewer_Jutm · 2026-05-18
**Good paper with interesting theory and thorough evaluation**

**Rating:** 8
**Confidence:** 3

**Review:**

# Summary

The authors consider solving streaming problems by querying tabular foundation models using a context set of previous labeled examples. They propose a novel theoretically-grounded algorithm for admitting/evicting datapoints to/from a fixed-size context in a data-dependent manner (vs. previous work which uses a data-independent heuristic).

# Strengths

- I found this paper quite interesting and well-motivated.
- The authors released their code, and at a quick glance it appears formatted in a way which should be easy to use by others.
- The mathematical framing of their method is mostly clear, and helps solidify what they are doing and why.
- While their method is somewhat complicated, the design decisions are clearly explained, well-motivated, and justified by ablations.

# Areas for improvement

- Assumption A.5 seems to imply that labels are nearly a deterministic function of the input, i.e. if we observe the label of $x$ then we can confidently predict the labels of nearby $x'$. I can think of many realistic scenarios where this isn't approximately true -- e.g. if there is noise or are unobserved variables in the data-generating process.
- The main paper claims that "Theorem A.6 lower bounds this contribution by prediction-time uncertainty $H(Y_{x_t} \mid x_t, D_t)$", but the theorem doesn't directly do this -- it bounds a different term by $\alpha_t(z_t) (H(Y_{x_t} \mid x_t, D_t) - \delta - \epsilon)$ where both $\delta$ and $\epsilon$ also appear to be functions of $z_t$, from assumptions A.4 and A.5. This seems to be a fairly large gap, and it's not obvious to me why the theorem motivates your algorithm here.
- I don't see it stated anywhere what the val set is which is used for hyperparameter tuning. For streaming models it seems important to have a val set sampled from times strictly before the test set to avoid leaking the particular type of distribution shift.
- It seems like the only performance metric is accuracy, even though some of the datasets are "highly" or "extremely" imbalanced. Would probably be better to use balanced accuracy or something.

# Detailed comments

- Would be nice to explicitly state that $I$ is a pointwise mutual information w.r.t. $D_t$ in Eqn. 2. This threw me off on first reading because I read it as normal mutual information.
- It seems like the proposed method would require recomputation of the entire KV cache for the context dataset whenever it is changed. Although this might be unavoidable for methods with dynamic context sets. I would be interested to hear the authors' thoughts on this.

# Justification of score

I enjoyed reading this paper, and liked their mathematical framing, though it appears there might be some gaps or loose bounds (unless I'm missing something). The ablation studies and evaluations appear thorough. It also sets up their problem and desiderata in a very clean way which may be useful to other work in the area, independently of the specifics of their method.

---

### Official Review · Reviewer_Jm8F · 2026-05-22
**Review of CURE**

**Rating:** 9
**Confidence:** 4

**Review:**

## Summary
This paper proposes CURE, a context management policy for applying tabular foundation models (TFMs) to stream learning. Rather than updating model weights to handle distribution shift (as classical methods do), CURE manages which labeled examples are retained in the TFM's context window. It introduces a "future-information view" that motivates three design signals: recency, uncertainty, and redundancy. Experiments across seven streams show consistent improvements over classical baselines and the prior DualFIFO method.

## Strengths
### 1. Theoretical motivation:
The future-information view is a principled framework that cleanly decomposes into three actionable signals, giving the method a coherent justification rather than being purely heuristic.
### 2. Backbone agnosticism:
Evaluating across four TFM backbones meaningfully supports the claim that CURE is a transferable interface rather than a backbone-specific trick.
### 3. Strong empirical results:
CURE achieves first place on all seven streams against classical baselines, with gains up to +19.59 points, and improves over DualFIFO in 17 of 18 backbone comparisons, demonstrating both absolute and relative strength.

## Weakness:
### 1. Gap between theory and implementation:
The theoretical choices rely on assumptions (local entropy consistency etc.) that are stated but never empirically verified or discussed qualitatively on the actual benchmark datasets. It is unclear how well these assumptions hold in practice.

## Questions for Authors:
1. Including empirical evidence that the theoretical results hold in actual benchmark datasets would be interesting to observe.

---

### Official Review · Reviewer_DTnq · 2026-05-22

**Rating:** 7
**Confidence:** 3

**Review:**

This paper proposes a context management strategy for tabular FMs in stream learning. They reframe adaptation as a problem of managing the in-context examples instead of making parameter updates.

The proposed strategy is well motivated through an information-theoretic perspective, and is easy to follow. The authors have evaluated their approach on six real datasets, classical baselines and tabular FMs. The results are promising.

The main contribution of the paper is in using existing well known ideas and applying them to tabular FMs for stream learning.